# Full-Field Bridge Deflection Monitoring with Off-Axis Digital Image Correlation

**DOI:** 10.3390/s21155058

**Published:** 2021-07-26

**Authors:** Long Tian, Jianhui Zhao, Bing Pan, Zhaoyang Wang

**Affiliations:** 1School of Science, China University of Geosciences, Beijing 100083, China; 2019200002@cugb.edu.cn; 2Institute of Solid Mechanics, Beihang University, Beijing 100191, China; panb@buaa.edu.cn; 3Department of Mechanical Engineering, The Catholic University of America, Washington, DC 20064, USA; wangz@cua.edu

**Keywords:** off-axis DIC, full-field determination, deflection measurement, initial parameter estimation

## Abstract

Video deflectometer based on using off-axis digital image correlation (DIC) has emerged as a robust non-contact optical tool for deflection measurements of bridges. In practice, a video deflectometer often needs to measure the deflections at multiple positions of the bridge. The existing 2D-DIC-based measurement methods usually use a laser rangefinder to measure the distance from each point to the camera to obtain the scale factor for the point. It is only suitable for the deflection measurements of a few points since manually measuring distances for a large number of points is time consuming and impractical. In this paper, a novel method for full-field bridge deflection measurement based on off-axis DIC is proposed. Because the bridge is usually a slender structure and the region of interest on the bridge is often a narrow band, the new approach can determine the scale factors of all the points of interest with a spatial straight-line fitting scheme. Moreover, the proposed technique employs reliability-guided processing and a fast initial parameter estimation strategy for real-time and accurate image-matching analysis. An indoor cantilever beam experiment verified the accuracy of the proposed approach, and a field test of a high-speed railway bridge demonstrated the robustness and practicability of the technique.

## 1. Introduction

The deflection or vertical displacement of a civil infrastructure under load is an important indicator that reflects the mechanical behavior of the structure. The deflections induced by various static or dynamic loads can provide essential information for reliability and safety assessment. Therefore, it has become a critical parameter for structural health monitoring in many civil engineering structures such as bridges and roadbeds. In literature, both contact and non-contact sensors or techniques are widely used for measuring displacement responses of engineering infrastructures. For instance, the linear variable differential transformer (LVDT) is a highly sensitive transducer for displacement measurement in a specific direction [1,2,3]. As it requires solid base support during measurements, its applications are highly limited. Alternatively, non-contact approaches, such as Global Positioning System (GPS) [4,5], microwave radar interferometry [6,7], and computer vision-based optical techniques [8,9,10,11,12,13,14,15], have been employed to overcome the drawbacks of the conventional contact sensors or methods. The GPS sensors are easy to use and can measure displacements in all directions, but the measurement errors in the vertical direction are larger than those in other directions [16,17,18]. The microwave radar interferometry system allows remote displacement measurement with a fair resolution, however, it typically requires mounting reflection components on the structure. Among those non-contact techniques, the computer vision-based optical methods seem to be a more practical and promising approach due to their quick, simple, and automatic measurement capabilities. Most existing vision-based optical methods for infrastructure deflection detections rely on tracking a customized reference target attached to the region of interest on the structure.

In recent years, with the rapid development of digital imaging equipment and image processing algorithms, bridge deflection measurement methods based on the principles of computer vision have continued emerging [19,20]. Digital image correlation (DIC) is a representative of such techniques and it is widely employed in the experimental mechanics community. The basic idea of the DIC technique is to detect the differences between two images of the same target captured before and after deformation with subpixel accuracy. Many researchers have used the technique for experimental characterizations of material properties and structural behaviors [21,22]. As DIC technique becomes prevalent, many attempts have been made to apply it to remote displacement measurements. In particular, video deflectometers have been used in quite a few engineering practices. For example, a vision-based system for multi-point structural dynamic displacement measurements has been developed [23]. Feng et al. [24] developed a targetless non-contact measurement system to characterize the dynamic displacements of railway bridges. Xu et al. [8] utilized a binocular vision setup to measure the deck deformation and cable vibration of a cable-stayed footbridge under pedestrian loading. In general, vision-based techniques measure the structural displacements by first tracking the pixel displacements in two images and then transforming the image displacements into physical displacements of the corresponding points on the object surface through coordinate transformation. To facilitate such a transformation, the camera’s optical axis should be perpendicular to the object’s surface. In such a case, the coordinate transformation between the physical displacement on the target and the image displacement in the images can be substantially simplified as a linear relation with a constant scale factor. For this reason, the vision system in the aforementioned applications should be placed roughly in the same horizontal plane as the target. As indicated by Feng et al. [24], if the vision system does not horizontally face the target object, the effect of the inclined pitch angle must be considered for reliable displacement detections. However, in many situations (such as outdoor field tests), it is often unavoidable to tilt the optical axis of the camera by a certain angle to track the region of interest on the target surface. In this paper, such a situation is coined as off-axis measurement and used hereafter.

Many researchers have recently carried out relevant work in response to the inevitable off-axis measurement problem in practice. Pan et al. [12] proposed an imaging model which allows off-axis measurement of vertical displacement by capturing images from a far distance. They attached a camera to a theodolite and aligned it with the telescope on the theodolite to determine the orientation of the camera’s optical axis. By measuring the distance from the system to the target with a laser rangefinder, the vertical displacement of the target can be calculated. Wang et al. [25] extended Pan’s work. They proposed considering the variations of pitch angles for different points on the target to improve the measurement accuracy if the target structure is large. Furthermore, Yu et al. [26] utilized smartphone gyroscope data to determine the rotation matrix between the defined world coordinate system and the camera coordinate system. It is noteworthy that most off-axis measurement methods can only handle a limited number of points and cannot obtain the accurate deflection information of the entire bridge. This limitation hampers its measurement capability in practical applications.

Aiming at the problem of off-axis full-field deflection measurement, this paper first introduces the principle of the full-field bridge deflection measurement method based on using an off-axis DIC scheme. On this basis, the algorithm of calculating full-field scale factors for the bridge is described, followed by proposing a suitable initial parameter transfer strategy for real-time full-field image matching. Finally, an indoor static load experiment and an on-site bridge measurement are presented to validate the accuracy and effectiveness of the proposed full-field bridge deflection measurement method.

## 2. Principle of Full-Field Bridge Deflection Measurement

Figure 1 shows the video deflectometer employed in this paper for full-field bridge deflection measurement. The video deflectometer consists of a high-speed video camera (Daheng Imaging, Mer-131-210u3m, 8-bit gray image, image resolution: 1280 × 1024 pixels, frame rate: 210 fps), a fixed-focus lens (the focal length can be selected according to the actual imaging needs, and the F-number of the lens can be set to a large value, such as f/16, to obtain a small aperture size with a large depth of field), a laser rangefinder (Bosch GLM200, maximum distance: 200 m, measurement accuracy: ±1 mm), an electronic theodolite, a tripod, and a laptop. It is important to point out that although it is desired to use a camera with ultra-high resolution and ultra-high capturing speed, there is a trade-off between resolution and speed (or sampling rate), in addition to the cost concern in practice. For this reason, the aforementioned camera, with a good balance of resolution and speed, is adopted, which can satisfy the measurement requirement. In the system, the video camera is fixed on the platform of the electronic theodolite. The pitch angle of the camera can be adjusted according to the height of the field of interest. The video camera is connected to a laptop, via a USB connection, to capture the video or images of the target bridge.

Due to the constraints in the field test, there is usually a pitch angle between the camera axis and the horizontal plane when imaging a bridge, as shown in Figure 2. In such an off-axis imaging situation, the distance from each measurement point to the camera along the optical axis differs, resulting in different magnifications for different points. Moreover, the non-perpendicularity between the camera’s optical axis and the bridge side plane also contributes to magnification variations. Therefore, it is essential to calculate the scale factor (SF) of each point of interest to convert its image displacement into its physical deflection. For instance, if the SF at a specific point is 10 mm/pixel and the maximum error of the DIC algorithm is about 0.01 pixel, then the maximum error of the measured deflection is about 0.01 pixel × 10 mm/pixel = 0.1 mm. Such an accuracy estimation should be performed before actual measurements. For the deflection measurement of multiple discrete points on the bridge, the detailed principle and procedure can be found in our previous research [12]. This work focuses on the full-field deflection measurement of the bridge. As illustrated in Figure 2, the region of interest (ROI) is the area that includes most of the side of the bridge span. Since the ROI is typically long and thin, it can be selected as a thick line. Accordingly, it is an essential assumption in the proposed work that all the points in the ROI of the bridge span are on a spatial line. That is, the points in the ROI are basically all the discrete points distributed on the line. It is worth noting that the ROI should be selected along the line with the most distinctive features of intensity change. Because the mean intensity gradient [27] of such an ROI is large, it would lead to the highest possible accuracy of measurement. With the existing single-point deflection measurement method, detecting the deflections of every point on the measurement line requires at least hundreds of times of distance measurements with a laser rangefinder, which is evidently impractical.

To conduct a full-field bridge deflection measurement, we first propose a full-field scale factor determination method assisted by a laser rangefinder. The laser rangefinder helps calculate the scale factors of a few (three or more) points by measuring their distances to the camera. Afterward, a fast initial parameter transfer strategy is employed for rapid 2D-DIC analysis. As illustrated in Figure 2, the full-field bridge deflection measurement based on the off-axis DIC analysis consists of the following five steps:Measurement preparation. Choose a fixed-focus lens with a focal length suitable for the measurement, set up the video deflectometer at a convenient location, align the camera, as well as adjust the lens focus and aperture to capture desired images of the bridge.Full-field scale factor determination. Specify the ROI line on the bridge, use the laser rangefinder to measure the distances from several points (no less than three) on the bridge to the camera, identify the pixel coordinates of these points in the camera view, and then fit them into the spatial straight-line equation of the bridge. According to the geometric relationship, the distances from every measurement point on the ROI line to the camera can be derived. The full-field scale factors of the ROI points can then be calculated.Image acquisition. Capture a video or a sequence of images of the bridge during the loading process.Image displacement calculation. Use the 2D-DIC imaging matching analysis to calculate the image displacement of each pixel corresponding to the ROI point on the bridge.Deflection calculation. Convert the image displacement of each point on the ROI line into its physical deflection with its scale factor.

In the above procedure, the full-field scale factor determination and image displacement calculation of each measurement point on the ROI line are the key to the full-field bridge deflection measurement. These two essential steps are elaborated in the following sections.

## 3. Full-Field Scale Factor Determination

The distances from different measurement points on the bridge to the camera along the optical axis direction are usually different, so the scale factor generally varies from point to point. In order to determine the specific scale factor for each individual point on the ROI line of the bridge, an off-axis imaging model is established based on the principle of the pinhole imaging model. Figure 3 shows the imaging geometries, where it is assumed that the bridge is subjected to a downward load. The load causes a displacement of a point $P_{1}$ on the bridge to point $P_{2}$ with a vertical displacement V and a movement of the corresponding pixel $p_{1}\left(x,y\right)$ in the captured image to pixel $p_{2}\left(\;x^{\prime },y^{\prime }\right)$ with a displacement v.

The governing equation of the deflection for a point on the bridge can be expressed as [11]:(1)$$\begin{array}{c} V=k_{SF}v\\ k_{SF}=\frac{L}{\sqrt{\left[{\left(x-x_{c}\right)}^{2}+{\left(y-y_{c}\right)}^{2}\right]l_{ps}^{2}+f^{2}}}\frac{l_{ps}}{cos\beta } \end{array}$$
where $k_{SF}$ is the scale factor of point $P_{1}$, β is the pitch angle of the camera orientation, $l_{ps}$ is the physical size of each CCD sensor pixel, f is the focal length, and L is the distance from the measurement point on the bridge to the camera. In general, $f\gg \sqrt{\left[{\left(x-x_{c}\right)}^{2}+{\left(y-y_{c}\right)}^{2}\right]}l_{ps}$, thus the scale factor can be simplified as follows:(2)$$k_{SF}\approx \frac{L}{fcos\beta }\cdot l_{ps}$$

In Equation (2), f, β, and $l_{ps}$ are constant for the system, whereas L is distinct for different measurement points. Consequently, *L* plays a critical role in determining the scale factor. Although *L* can be measured using a laser rangefinder, it is impractical to apply the measurement to every point on the ROI line. Instead, *L* for nearly all the ROI points must be determined automatically.

To illustrate the full-field scale factor determination method, we first establish a camera coordinate system $x_{c}y_{c}z_{c}$ and an image coordinate system xy, as shown in Figure 4. At least three points on the bridge are selected to measure their distances to the camera to fit the spatial straight-line equation of the bridge. Denoting the world coordinates of the selected points as $P_{1},P_{2},\ldots ,P_{n}$, their image coordinates as $p_{1},p_{2},\ldots ,p_{n}$, and the corresponding measured distances as $L_{1},L_{2},\ldots ,L_{n}$, because $p_{i}$ can be converted to its camera coordinates $p_{i}^{\prime }$ with $p_{i}^{\prime }=\left(x_{i}l_{ps},y_{i}l_{ps},f\right)$, the magnification $M_{i}$ of point $P_{i}$ can be defined as:(3)$$M_{i}=\frac{L_{i}}{\sqrt{\left[x_{i}^{2}+y_{i}^{2}\right]l_{ps}^{2}+f^{2}}},\;i=1,2,\ldots ,n$$

According to the homothetic triangle theory, point $P_{i}$ can be associated with its image point $p_{i}$ as:(4)$$P_{i}=p_{i}^{\prime }\cdot M_{i},i=1,2,\ldots ,n$$

In order to obtain the magnification of every point on the ROI line, the spatial straight-line equation can be fitted under the camera coordinate. First, the spatial straight-line equation of the bridge can be defined as:(5)$$\frac{x-{\hat{x}}_{b}}{d_{1}}=\frac{y-{\hat{y}}_{b}}{d_{2}}=\frac{z-{\hat{z}}_{b}}{d_{3}}$$
where $\left({\hat{x}}_{b},{\hat{y}}_{b},{\hat{z}}_{b}\right)$ are the mean coordinates of the multiple points whose distances to the camera have been measured, and $\left(d_{1},d_{2},d_{3}\right)$ is the direction vector. $\left(d_{1},d_{2},d_{3}\right)$ can be estimated by the singular value decomposition (SVD). Specifically, after performing the SVD on the matrix formed by the standardized coordinates of all points, the left singular vector corresponding to the largest singular value is the direction vector.

After getting the spatial straight-line equation of the bridge, the distance from any ROI point on the bridge to the camera can be easily obtained. Given that a point Q on the bridge can be expressed as:(6)$$Q=\left(xl_{ps},yl_{ps},f\right)\cdot M$$

Substituting it into Equation (5), the magnification M can be calculated. Then the camera coordinates of point *Q*, $\left(Q_{x},Q_{y},Q_{z}\right)$, can be obtained. Subsequently, distance L from point Q to the camera can be obtained as:(7)$$L=\left|OQ\right|=\sqrt{Q_{x}^{2}+Q_{y}^{2}+Q_{z}^{2}}$$

Finally, following Equation (2), the scale factor can be determined. Applying this process to each measurement point on the ROI line of the bridge automatically yields the required full-field scale factors.

## 4. Full-Field Image Displacement Calculation with DIC

The full-field image displacement calculation with the DIC technique consists of the following primary steps: (1) determining the subset size for the analysis, (2) selecting proper correlation function and shape function, and (3) tracking the displacements of pixels in the undeformed (reference) and deformed images. Because local strain variations around the measurement points for the bridge deflection measurement at a far distance are negligible, the analysis can use a zero-order shape function that only considers rigid body translation. In addition, a robust cost function named zero-mean normalized sum of squared difference (ZNSSD) is adopted in the analysis, which is expressed as:(8)$$C_{\mathrm{ZNSSD}}\left(\Delta p\right)=\underset{\xi }{\sum }{\left\{\frac{\left[f\left(x+W\left(\xi ;\Delta p_{0}\right)\right)-f_{m}\right]}{\sqrt{\sum_{\xi }{\left[f\left(x+W\left(\xi ;\Delta p_{0}\right)\right)-f_{m}\right]}^{2}}}-\frac{\left[g\left(x+W\left(\xi ;p_{0}\right)\right)-g_{m}\right]}{\sqrt{\sum_{\xi }{\left[g\left(x+W\left(\xi ;p_{0}\right)\right)-g_{m}\right]}^{2}}}\right\}}^{2}$$

In Equation (8), f(x) and g(x) are the grayscale intensities of the pixel x within the reference and target subsets, respectively. $f_{m}$ and $g_{m}$ are their mean of the subset. $\xi ={\left(\Delta x,\Delta y\right)}^{T}$ denotes the *x*- and *y*-directional displacement components of the reference subset center, $W\left(\xi ;p_{0}\right)$ is the mapping function of each pixel point in deformed image subset, $W\left(\xi ;\Delta p_{0}\right)$ is the incremental shape function acting on the reference image subset, and $p_{0}={\left(u,v\right)}^{T}$ indicates the parameters of the zero-order shape function.

The sub-pixel displacement of each reference subset center can be obtained by optimizing Equation (8) with the inverse compositional Gauss–Newton algorithm (IC-GN) and bicubic spline interpolation method. Although the IC-GN algorithm is well known for being highly accurate and efficient, the algorithm will not converge well without an accurate initial estimate of the shape function parameters. That is, a good initial parameter estimate is essential for the proposed full-field bridge image displacement measurement. In this work, two initial parameter transfer strategies are adopted: (1) reliability-guided DIC (RG-DIC) scheme, and (2) initial parameter transfer scheme with motion estimation.

With RG-DIC, a few points with high mean intensity gradients are first selected as seeds from the ROI, and a fine search routine carries out the initial displacement estimations of these seed points in the reference and deformed images (i.e., the first two frames of images). After tracking the seed points, the initial displacements of the remaining points can be obtained from their neighbor points according to the RG-DIC scheme. For the subsequent images, the initial displacement value of each point directly adopts the displacement determined in the previous image. By sequentially transferring the initial displacements with motion estimation, the required initial estimates for new point analysis can be acquired reliably and efficiently. In most cases, this approach is quite robust and effective. Only in the case of point tracking failure, the initial estimate needs to be searched again within the image. The two initial parameter transfer strategies are showed as Figure 5.

### 4.1. Reliability-Guided Initial Displacement Value Transfer Strategy

The RG-DIC method is mainly used to obtain the initial parameter or displacement estimates of the points in the first deformed image. In the conventional initial parameter transfer process, the transfer scanning direction is often row-by-row or column-by-column, which is prone to error propagation and subsequently may lead to substantially longer processing time or cause wrong analysis results. The RG-DIC method [28] can immensely improve the accuracy of initial parameter transfer (or initial displacement transfer since the parameters are the horizontal and vertical displacements for a zero-order shape function). Since the ROI of the bridge is typically a straight line, a modified RG-DIC approach is proposed here (Figure 6):Select seed points on the ROI line according to the mean intensity gradients of the reference image;Sort the seed points following the ascending order of the correlation coefficients and add them to a queue Q;Remove the element from the front of the queue and transfer its value to uncalculated neighbor points on both sides. In the meantime, insert each new calculation point into Q at the correct position based on its correlation coefficient;Repeat step (iii) until Q is empty, indicating that all the points in the ROI have been successfully analyzed.

With this method, the point with the lowest correlation coefficient has the priority to transfer its analyzed results to its neighbor points as initial estimates, and these points are to be processed next since they have the most reliable initial estimates. In contrast, if an analyzed point has a high correlation coefficient, the calculation of its neighbor points will be deferred. Accordingly, the erroneous transfer of initial displacement value can be avoided to the utmost extent, especially for the bridge measurement where desired texture intensity variations may not be available.

### 4.2. Initial Estimate Transfer with Motion Estimation

Real-time, full-field deflection measurement requires fast calculation. The RG-DIC method helps improve the reliability and speed of the initial estimates for the first deformed image. For subsequent deformed images, the following initial estimate transfer with motion estimation can be generally more efficient. It is noted, however, that the initial estimate transfer scheme is not always effective during the tracking of every point. If the correlation analysis of some points fails, their initial estimates must be searched again within the domain of the bridge image.

In real-time deflection measurement, the time interval between two adjacent frames is short. Thus, the displacement of a point in three adjacent image frames can be approximately treated as a linear function with respect to time. Denoting the displacements of a point in the (*n* − 2)-th and (*n* − 1)-th image frames as $\left(u_{n-2},\;v_{n-2}\right)$ and $\left(u_{n-1},\;v_{n-1}\right)$, respectively, the displacement of the same point in the *n*-th frame, $\left(u_{n},\;v_{n}\right)$, can be approximated as:(9)$$\{\begin{array}{c} u_{n}=2u_{n-1}-u_{n-2}\\ v_{n}=2v_{n-1}-v_{n-2} \end{array}$$

Equation (9) is based on linear motion within a short time interval. Such an initial estimate transfer with motion estimation allows quick and reliable image analysis, which is important for the proposed real-time deflection measurement.

## 5. Experimental Verification

To verify the effectiveness and accuracy of the proposed full-field bridge deflection measurement technique, we designed a comparative experiment of a thin cantilever beam subjected to static loads. The cantilever not only morphologically simulates the bridge well but also facilitates the determination of ground-truth displacements.

Figure 7 shows the cantilever beam and the schematic setup. The 950 mm long steel beam is fixed on the left end and is free on the other end with static loading. Artificial speckles are sprayed on the front side of the beam to provide sufficient texture variation for easy and reliable image correlation purposes. Furthermore, five dial digimatic indicators are positioned at different locations of the beam to acquire the true deflections, which serve as the ground-truth reference. Specifically, the horizontal distances of the five locations to the loading position are 1.77 m, 1.70 m, 1.61 m, 1.50 m, and 1.43 m, respectively. The focal length of the lens is 8 mm, and the pitch angle of the deflectometer is 11°. In the image coordinate system, the measurement line of interest satisfies the following spatial straight-line equation:
(10)$$\frac{x+110.58}{0.8291}=\frac{y+43.92}{0.0689}=-\frac{z-1589.67}{0.5549}$$

With the proposed approach, the distance of every point on the line of interest can be determined and the corresponding scale factor can then be obtained according to Equation (2). Figure 8 shows the detected scale factors of the beam. With the full-field scale factors, the image displacement can be transformed into beam deflection succinctly.

During the experiment, five different weights, 1.0 kg, 3.0 kg, 3.5 kg, 4.0 kg, and 4.5 kg, were loaded and unloaded on the cantilever beam. The full-field deflection curves of the beam under the loading phase are shown in Figure 9a. In general, the full-field deflection curves of the beam are smooth, and the deflections at the free end vary notably and demonstrate to be proportional to the magnitudes of the loads. To assess the measurement accuracy, Figure 9b shows the deflections detected by the dial indicator and the results provided by the proposed technique at 11 loading stages. Table 1 summarizes the mean errors and standard deviations of all the loading stages. It can be seen from the table that the measurement accuracy can reach 0.085 mm for the experiment.

## 6. Application of Deflection Measurement

To evaluate the practicability of the proposed full-field bridge deflection measurement, a field experiment has been carried out on a high-speed railway bridge on the south side of Jingtai Bridge in Beijing. The bridge is located on the line from Beijing Railway Station to Tianjin Railway Station, and it has a main span of about 16 m in the center section. The video deflectometer is placed on the sidewalk of a road running below the bridge. The distance from the camera to the bridge is around 30 m, which allows capturing the section of interest of the bridge. The camera is equipped with a 16 mm focal-length lens and has a pitch angle of 3.5°. Five points are selected on the side of the bridge, and their distances to the camera measured by the laser rangefinder are 33.07 m, 31.80 m, 30.72 m, 28.21 m, and 27.25 m, respectively. After identifying these points in the images, the spatial straight-line equation of the bridge is fitted as:(11)$$\frac{x+1265}{0.8170}=-\frac{y+1363}{0.0028}=-\frac{z-30013}{0.5766}$$

With Equation (11), the distance of every point on the ROI line can be calculated. Following that, the full-field scale factors of the bridge are determined, as shown in Figure 10. Finally, the full-field deflections can be calculated.

Figure 11b and Figure 12 show the full-field deflection curves during the high-speed train passing across the bridge, including five representative moments: before arrival, three selected moments when the train was over the bridge, and after departure. The results are consistent with the qualitative deflection changes of the bridge at different stages when a high-speed train passes across. It verifies that the proposed full-field bridge deflection measurement method is valid and can effectively detect the deflection of the bridge. For every measurement point on the bridge, a time-varying deflection curve can also be obtained, as shown in Figure 11b. Despite that, the field test reveals that it is challenging to get well correlated full-field image displacements with natural texture patterns in some regions on the side of the bridge. Consequently, some post-processing work, including mean filtering and median filtering, may be demanded to reduce the measurement noise. Other handlings, such as spraying artificial speckle patterns and attaching LED-illuminated speckle targets to the bridge for usage at night, can also help cope with the uncorrelation problem.

## 7. Conclusions

The deflection measurements of a bridge can provide important information for its reliability and safety assessment. A non-contact, robust yet simple technique for such measurements is highly demanded. This paper presents a real-time, non-contact, and full-field bridge deflection measurement method based on an off-axis DIC approach. The proposed technique involves two key components: determination of off-axis full-field scale factors and high-efficiency calculation of full-field image displacement. With the spatial straight-line fitting method, the full-field scale factors can be determined upon measuring the distances from a few selected points on the bridge to the camera. By applying the reliability-guided DIC analysis and the initial estimate transfer with motion estimation scheme, the full-field image displacement maps can be efficiently and accurately calculated. The indoor cantilever beam experiment and the on-site actual bridge deflection measurement have validated the robustness and effectiveness of the proposed approach.

Compared with the conventional single-point bridge deflection measurements, the proposed full-field bridge deflection measurement possesses the following advantages: (1) There is no need to measure the distance to the camera for every point of interest, only the distances to the camera from a selected three or more points are required. (2) The initial parameter estimation method for the 2D-DIC analysis is reliable and efficient. (3) The deflection curve of a whole bridge under external load can be accurately obtained.

The deflection measurement accuracy of the proposed technique depends on many factors, such as the image resolution, field of view, measurement distance, frame rate of the camera, and environmental constraints. The optimal conditions of these factors may not be available in every real-world application and, thus, the performance of the proposed technique could be more or less limited. Furthermore, a bridge in the real world may not always have the desired natural texture patterns available on its side surface for the proposed measurement. There could be some points and segments without sufficient intensity variations in the ROI for reliable DIC analysis in practice. We would have to skip the measurements of these points or make artificial feature patterns on the measurement surface through labeling or spraying. As a vision- and image-matching-based technique, the measurement accuracy highly depends on the image quality and the intensity variations in the ROI. Consequently, we consider that further studies should put emphasis on exploring novel approaches that can achieve image matching with high accuracy even when the images have small or minimal variations of feature intensities.

## Figures and Tables

**Figure 1 sensors-21-05058-f001:**
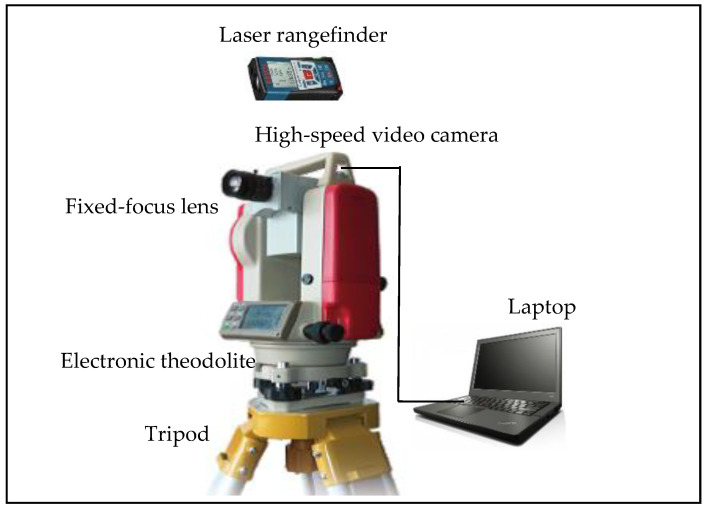
Video deflectometer for full-field bridge deflection measurement.

**Figure 2 sensors-21-05058-f002:**
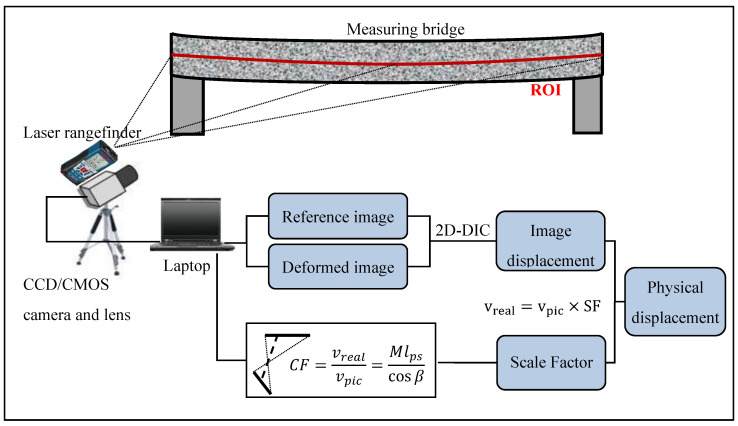
Schematic of full-field bridge deflection measurement.

**Figure 3 sensors-21-05058-f003:**
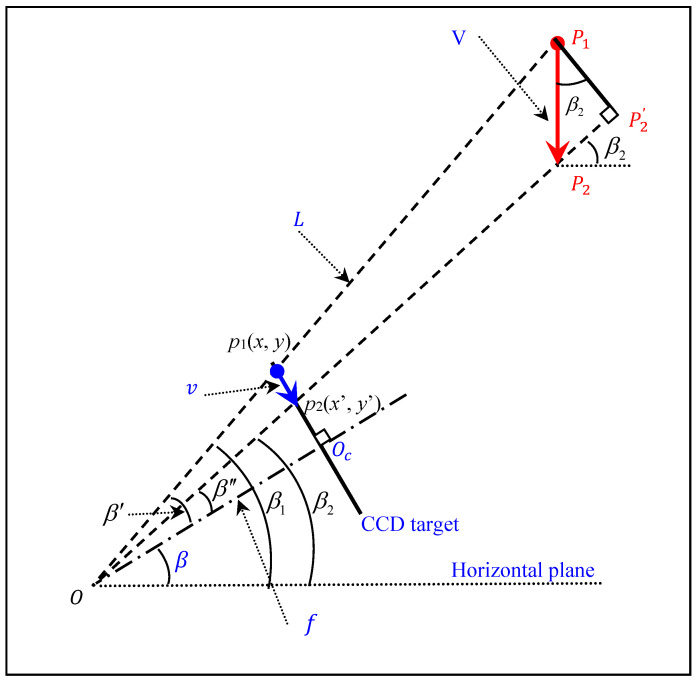
Schematic of the single-point deflection measurement.

**Figure 4 sensors-21-05058-f004:**
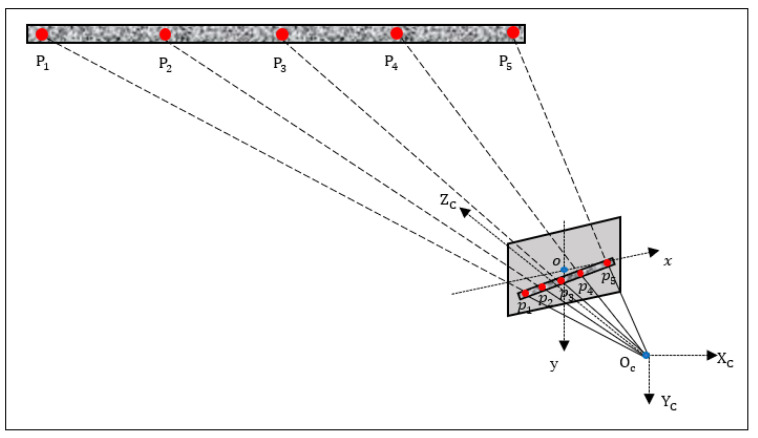
Schematic of the full-field scale factor determination, where $P_{1},P_{2},\ldots ,P_{5}$ are the points whose distances to the camera have been measured, and $p_{1},p_{2},\ldots ,p_{5}$ are the corresponding pixels in the captured image.

**Figure 5 sensors-21-05058-f005:**
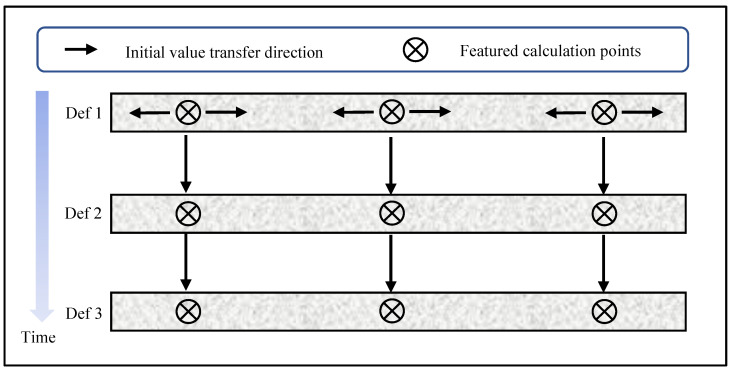
Illustration of initial displacement value transfer direction.

**Figure 6 sensors-21-05058-f006:**
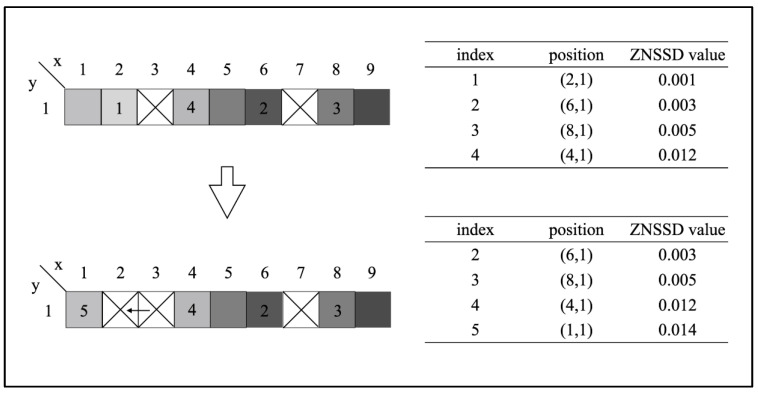
Schematic of modified RG-DIC based on ZNSSD.

**Figure 7 sensors-21-05058-f007:**
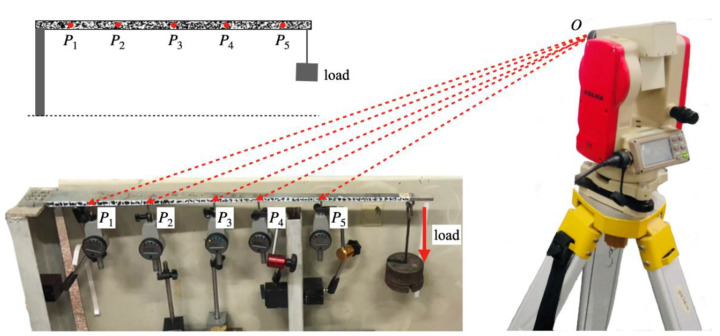
Measurement of a cantilever beam subjected to static loads.

**Figure 8 sensors-21-05058-f008:**
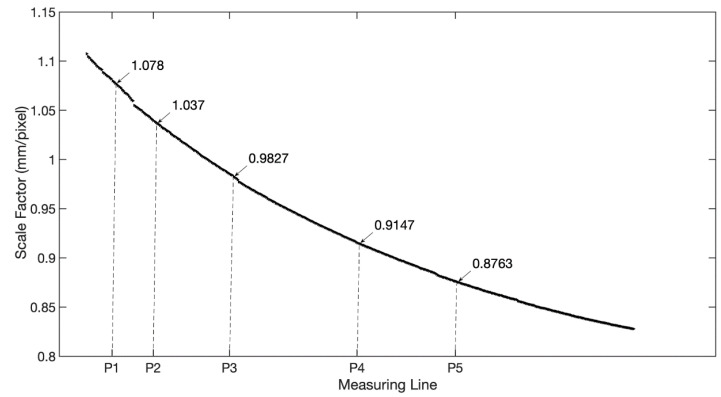
Full-field scale factors of the cantilever beam.

**Figure 9 sensors-21-05058-f009:**
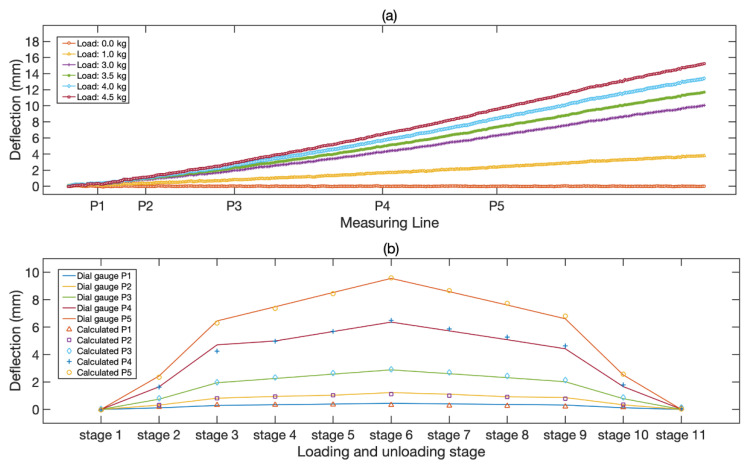
Experimental results of a cantilever beam under static loading: (**a**) full-field deflection curves of the beam, containing 6 different loads; (**b**) comparison of results obtained by dial indicator and video deflectometer, including 11 loading and unloading stages.

**Figure 10 sensors-21-05058-f010:**
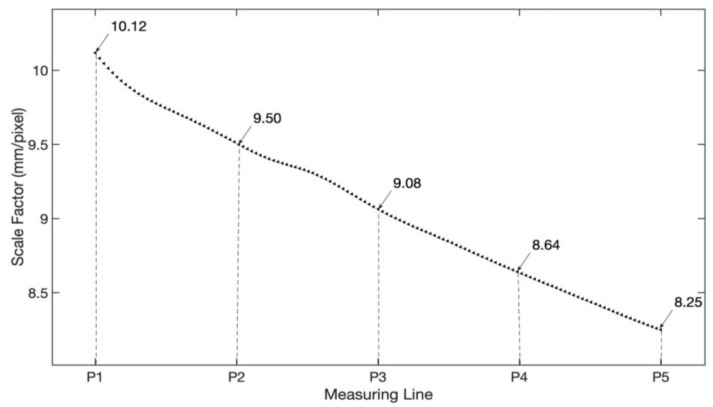
Full-field scale factors of the bridge.

**Figure 11 sensors-21-05058-f011:**
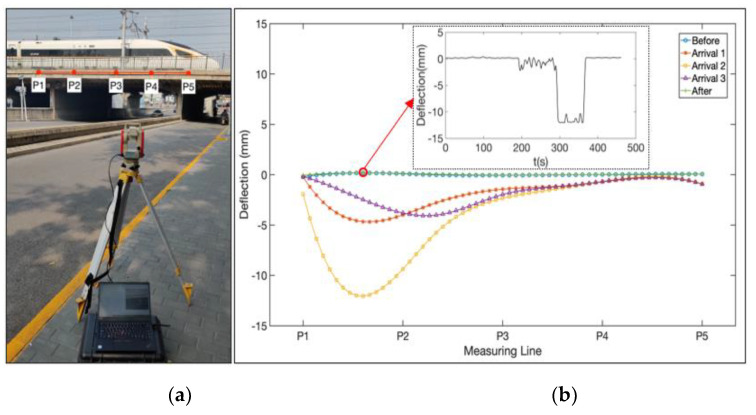
Field experiment: (**a**) setup of the field experiment, (**b**) full-field deflection curves of the bridge that includes five representative moments: before arrival, three selected moments after arrival, and after departure.

**Figure 12 sensors-21-05058-f012:**
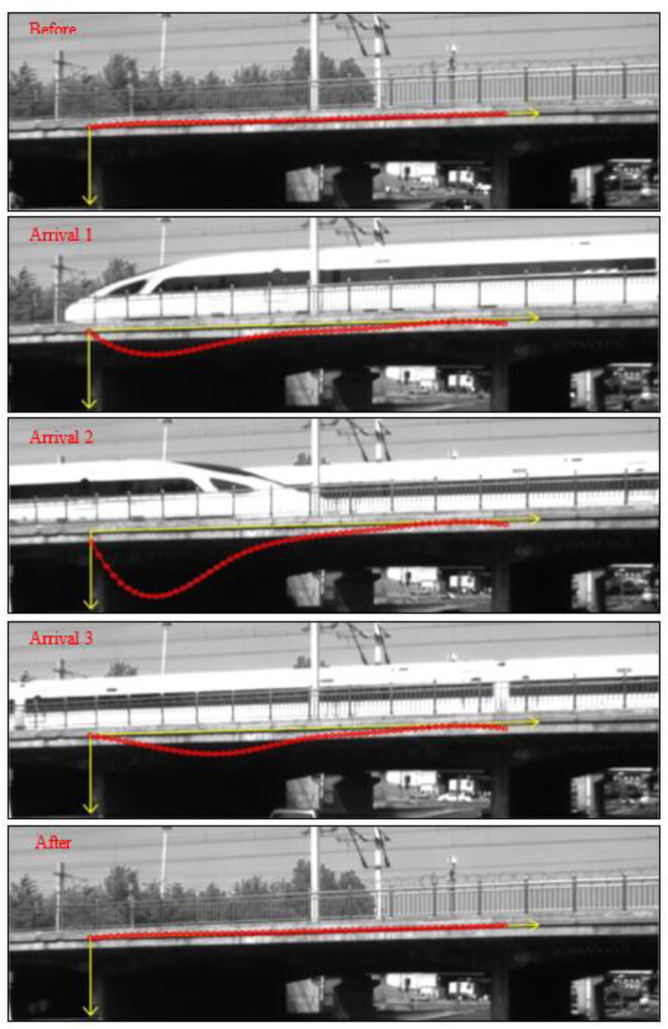
Full-field bridge deflection curves at five representative moments.

**Table 1 sensors-21-05058-t001:** The mean errors and standard deviations at five measurement locations.

Measuring Point	Mean Error (mm)	Standard Deviation (mm)
$P_{1}$	0.030	0.084
$P_{2}$	0.026	0.049
$P_{3}$	−0.085	0.041
$P_{4}$	−0.031	0.180
$P_{5}$	−0.005	0.110

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
