# Peer review of "Full-Field Bridge Deflection Monitoring with Off-Axis Digital Image Correlation"

_sensors, 2021, doi:10.3390/s21155058_

Round 1

Reviewer 1 Report

1) Avoid using hyphens to complete the word at the end of the line e.g. "measure-ment". Use justified text to fit everything between the margins.

2) More information about the section of interest of the on-site bridge is required e.g. Is the section acting as a cantilever? What kind of bridge is this? How will the approach change when applying to another type of bridge?

3) Please state all the assumptions while applying the technique to the real-world problems

4) What are some of the limitations of this technique? How accurate the calculations are and what is the measure of accuracy?

5) Please re-write the conclusions to reflect the research question and the objectives.

Reviewer 2 Report

The paper presents a bridge deflection measurement method based on an off-axis DIC approach.

The work is interesting, well organized, but some issues need to be fixed.

1 Introduction

  • I suggest that you describe DIC basis, considering some articles published by this journal.For example:

https://www.mdpi.com/1424-8220/20/19/5622

https://www.mdpi.com/1424-8220/21/4/1154

2 Principle of full-field bridge deflection measurement

  • In this paper, the determination of a ROI represents the main contribution to the measurement uncertainty.Such aspect should be addressed.
  • The resolution of the used image acquisition system is a parameter that should be discussed when analyzing large samples such as bridges.

4 Full-field image displacement calculation with DIC

  • Figure 5 is not cited in the text.

5 Experimental verification

  • The values, reported in table 1, should be defined up to 0.001 mm. A higher measurement resolution is unreal for your setup.

6 Application of deflection measurement

  • One of the main issues of DIC technique is the speckle pattern of the analyzed sample.In fact, often, improper natural speckles involve considerable measurement uncertainty or do not make measurement possible.In my opinion, the natural surface of a bridge is a typical case. This aspect should be considered in your measurements and should be discussed in your results.

Reviewer 3 Report

This paper presents a full-field bridge deflection monitoring method by using off-axis digital image correlation. A new approach is proposed to determine the scale factors of all points of interest (POIs), which are approximately aligned in a line. Therefore, there is no need to measured the distances between the camera and all POIs, which facilitates the practical application. Moreover, both laboratory and field tests are conducted to verify the measurement accuracy and the practical applicability of the proposed method. The topic of this paper is important and interesting to the readers in the field of structural health monitoring. Overall, the paper is clearly written. It is recommended for the publication given that the following concerns from this reviewer are addressed.

  • In order to get the accurate full-field bridge deflection, the picture of all POIs of the bridge should be in focus. It requires that the distances between the POIs on the bridge side and the camera cannot variate significantly. The authors should add some comments to address this issue.
  • Generally, the camera lens causes the distortion of the picture particularly near the edge of the picture. Will the distortion affect the accuracy of displacement measurement in the full-field bridge deflection monitoring?
  • A lot of paragraphs in the paper are not correctly aligned. For example, 2nd
  • The expression “Because the region of interest on the bridge is often a narrow band” in the abstract is kind of confusing, which needs rephrasing.

Round 2

Reviewer 1 Report

good job!

Reviewer 2 Report

All my comments have been fully and correctly addressed.

In my opinion, the manuscript can be considered for publication.